# Waist rotation angle as indicator of probable human collision-avoidance direction for autonomous mobile robots

Tatsuto Yamauchi⊚, Hideki Tamura ⓘ⊚*, Tetsuto Minami, Shigeki Nakauchi

Department of Computer Science and Engineering, Toyohashi University of Technology, Toyohashi, Aichi, Japan

⊚ These authors contributed equally to this work.
* tamura@cs.tut.ac.jp

## Abstract

The likelihood of pedestrians encountering autonomous mobile robots (AMRs) in smart cities is steadily increasing. While previous studies have explored human-to-human collision avoidance, the behavior of humans avoiding AMRs in direct, head-on scenarios remains underexplored. To address this gap, we conducted a psychophysical experiment to observe how humans react to an AMR approaching directly. The AMR was programmed to approach from various starting points, including a direct path toward participants, and their avoidance movements were recorded. Participants were instructed to evade by moving either right or left, with no strong preference for a particular direction observed. This suggests that avoidance direction is not strictly influenced by individual factors, such as adherence to regional traffic norms. Additionally, motion analysis revealed that participants instinctively twisted their waists in the direction of avoidance before evading. Further experiments assessed the role of waist rotation angle in influencing human comfort during AMR avoidance. The results indicated that early AMR avoidance improved participant comfort. Moreover, using an RGB camera allowed non-contact measuring without sensors, broadening the applicability of the technique. These findings suggest that waist rotation reliably predicts avoidance direction, and non-contact detection methods, such as RGB cameras, show substantial potential for broader applications. Further research will focus on improving the accuracy and robustness of these non-contact techniques.

## Introduction

In various regions worldwide, encounters with autonomous mobile robots (AMRs) on urban streets are becoming more frequent as governments and industries actively promote advancements toward smart cities [1–3]. AMRs are designed to follow planned delivery routes to optimize efficiency; however, they must ensure pedestrian safety, even in complex scenarios [4]. Consequently, numerous studies

**Data availability statement:** The data underlying the results presented in the study are available from the Open Science Framework repository (https://osf.io/ajymd/).

**Funding:** This paper is based on the results from project JPNP20004, subsidized by the New Energy and Industrial Technology Development Organization (NEDO). This work was supported by the CASIO SCIENCE PROMOTION FOUNDATION (39-55) funds, the Foundation of Public Interest of Tatematsu, and the 2021 Toyohashi University of Technology President Funding (Young Researchers). Additionally, general operational funds allocated to the laboratory by Toyohashi University of Technology may have indirectly contributed to this study through the provision of research facilities and equipment. There was no additional external funding received for this study.

**Competing interests:** The authors declare no competing interests.

in robotics engineering have explored various approaches to address this issue. For example, Kamezaki investigated control methods where a robot either guided or yielded to a human based on the estimated positional relationship between the two [5]. The study found that humans preferred a mutual concession approach to avoidance rather than one party yielding completely. These findings underscore the importance of incorporating human-like control methods in robots, which positively affect human perceptions. In other words, for AMRs to be effectively integrated into everyday life, their behavior should align with human expectations and preferences.

### Human-robot interaction

Several studies focus on achieving effective collaboration between humans and robots. In such scenarios, robots must analyze their environment by recognizing people and objects [6–8] and maintain an appropriate distance from humans [9,10]. The distance at which humans feel comfortable from a robot is generally greater than with another person [11]. If a robot encroaches on personal space, it can increase discomfort [12]. Moreover, people's perceptions of a robot, such as whether it appears artificial, can vary depending on the situation [13]. For instance, incorporating gaze information into a robot's design and programming it to make eye contact can enhance the feeling that the robot is a person rather than merely an artificial object [14,15]. This perception contributes to more effective human-robot teamwork [16]. Additionally, the acceptance of a robot's request can be influenced by its shape, modality, or method of communication [17]. A robot's facial expressions and demeanor can also affect the speed of human responses [18]. These findings highlight the significance of human perception in successful human-robot collaboration.

### Robot control using AI methods

For AMRs to be widely adopted in human environments, they must effectively navigate crowded spaces while fulfilling their objectives. To address this challenge, several studies focus on developing appropriate avoidance strategies using artificial intelligence methods. Research in this area includes developing obstacle-avoidance algorithms that incorporate human motion, intention, and preferences as part of the human model [19]. Additionally, algorithms have been created that enable robots to correct their paths and avoid obstacles if initial avoidance maneuvers are inadequate [20]. Dhouib developed the Dhouib-Matrix-SPP (DM-SPP) method, which rapidly solves the shortest path problem. When applied to AMRs, this method was compared with 12 conventional approaches and demonstrated superior accuracy and speed [21,22]. Furthermore, Chen et al. utilized deep learning techniques to focus on human traffic rules, particularly those related to prohibited behaviors, to create a system that allows robots to navigate autonomously in pedestrian-dense environments [23]. This research underscores the importance of incorporating human traffic rules and implicit walking behaviors into robot control systems.

## Cognitive science toward robots

The cognitive science perspective, which examines human obstacle-avoidance techniques, can be applied to understand how humans avoid collisions with robots. For example, humans use various strategies to navigate around static obstacles, such as barriers [24–26], poles [27], two poles [28,29], or larger and smaller interferers [30]. Additionally, moving people, who act as dynamic obstacles, often enter participants' surroundings, creating a dynamic environment [31,32]. Previous studies have explored how humans avoid dynamic obstacles approaching from orthogonal directions [33–36] or crossing their paths [37–41]. Several of these studies have replicated real-world scenarios in virtual reality [28,39–42]. Virtual reality experiments are advantageous because they eliminate potential physical collisions and increase the number of experimental variables, such as the presence of additional human walkers, which benefits participants.

## Avoidance behavior between humans and robots

Few cognitive science studies have examined human evasive actions against AMRs, which are expected to become common dynamic obstacles. Recent research has focused on understanding human behavior when interacting with AMRs. For instance, participants reported higher comfort levels when passing AMRs that mimic human avoidance paths or use pedestrian models, compared to those that do not [43,44]. Additionally, effective navigation interactions for humans are being studied in scenarios where AMRs are surrounded by crowds [45,46]. Neggers et al. investigated comfort in avoidance strategies when humans and AMRs cross paths orthogonally [47]. They found that AMRs exhibiting predictable behavior, such as maintaining a straight path or stopping, were perceived as more comfortable than those that changed direction. These findings highlight the benefits of studying human locomotion in conjunction with AMRs. However, existing cognitive science studies have primarily examined scenarios where humans cross AMRs orthogonally. Thus, the outcomes of imminent head-on collision scenarios—specifically, how humans choose an avoidance direction for safety—remain unclear.

## Aims of this study

This study aims to understand the strategies humans use to avoid AMRs approaching head-on through a psychophysical experiment. Specifically, we first investigate whether humans favor a particular side (right or left) when avoiding AMRs, estimated by a psychometric function analyzing the probabilities of rightward avoidance. We also examine how the avoidance direction is determined in such situations. Previous research has suggested that humans tend to move rightward to avoid collisions due to a cognitive bias linked to traffic rules in right-hand driving countries [39]. However, if this were the case, individuals in left-hand driving countries might exhibit different behavior. We hypothesize that humans may struggle to instinctively decide on an avoidance direction when faced with an unfamiliar object approaching them. Therefore, we conceptualized the human locomotor response to an inbound AMR by computing their avoidance probability with the AMR as the obstacle.

We hypothesize that humans predominantly avoid collisions by moving to a specific side when considering the AMR as either 1) a human or 2) an inorganic object. To test this, we used an actual AMR as the obstacle and recruited participants from a left-hand driving region. In Experiment 1, we implemented a technique to observe how humans evade an AMR approaching from five different directions, including head-on. A motion-tracking system was employed to estimate their walking trajectories, and we tested indices derived from human locomotion to predict the avoidance direction. In Experiment 2, we assessed human comfort with the anticipated avoidance of an AMR based on the waist rotation angle. Subsequently, in Experiment 3, we aimed to change the method of obtaining the waist rotation angle from a motion tracker to a camera. To achieve this, we utilized pose estimation with MediaPipe Pose and conducted the psychophysical experiment using the same procedure as in Experiment 2.

## Literature review

**Human AMR collision-avoidance behavior.** Human collision-avoidance behavior toward AMRs differs from that toward other humans. Vassallo et al. conducted an experiment in which humans and AMRs walked in the same

environment [48]. Participants were instructed to walk at their preferred speed from start to destination, knowing that the AMR might obstruct their path. The study found that humans tended to yield to the AMR even when they were initially in a position to pass first, unlike the interactions observed between two humans. Subsequently, the same research group programmed the AMR to implement a human-like collision-avoidance strategy, predicting which participant would pass first and adjusting its path accordingly. The results demonstrated that humans exhibited avoidance behavior toward the programmed AMR, similar to their behavior toward other humans [49]. These findings suggest that programming AMRs to mimic human behavior can facilitate smoother interactions between humans and robots.

However, these studies only addressed situations where the paths of humans and AMRs intersected orthogonally. Thus, the collision-avoidance behavior when pedestrians and AMRs face each other directly remains unclear. In a related study, Souza Silva et al. investigated how human avoidance strategies were influenced when encountering a human or a cylindrical object head-on [39]. Participants navigated around obstacles (either a cylinder or a virtual human) approaching from the right (+40°), left (-40°), or directly ahead (0°) while walking toward a distant target in a virtual reality environment. The results showed that for obstacles approaching from the sides, participants consistently avoided in the direction opposite to the approaching obstacle, regardless of whether it was a cylinder or a virtual human. When faced with a cylinder approaching head-on, the probability of avoiding to the right (55%) or to the left (45%) was nearly equal. In contrast, when confronted with a virtual human head-on, the probability of avoiding to the right increased to 75%. These results indicate that human avoidance behavior may vary depending on whether the obstacle is a human or an inanimate object. This variability suggests that human avoidance behavior might also differ when the obstacle is an AMR.

**Avoidance direction indicators.** For comfortable and safe passing, predicting avoidance using various indicators has been studied. Several studies have focused on the legs to predict avoidance behavior. Tomizawa and Shibata examined the landing position of the foot and studied the appropriate avoidance actions a robot should take to minimize collision risk when passing a pedestrian head-on [50]. They found that avoiding a pedestrian on the opposite side of the landing foot reduces the risk of collision. Tamura et al. developed a robot that could smoothly avoid pedestrians by predicting their movement direction, using a laser range finder to detect human legs [51].

Other research has focused on the gaze direction of pedestrians. Nummenmaa et al. explored how a person's gaze direction while walking affects eye movements and path choices [52]. They found that humans use the direction of an oncoming pedestrian's gaze to infer the pedestrian's movement direction and avoid directing their own attention in that direction. Jakobowsky et al. investigated whether gaze information from a robot influences a person's choice of avoidance direction [53]. They concluded that using gaze to communicate the direction of avoidance is effective, whether the counterpart is a robot or a person. Neggers et al. examined a strategy where the robot's gaze serves as a cue for predicting its behavior [54]. They suggested that robots should make eye contact and communicate their gaze direction to enhance human comfort.

However, some studies suggest that hands and feet alone do not significantly affect avoidance behavior. Fiset et al. investigated how local limb movements of pedestrians impact their planning and execution of avoidance and found that while local limb movements influence the minimum approach distance, they do not affect avoidance movements [55]. Lynch et al. studied whether the gaze of a virtual character influences participants' collision-avoidance behavior and reported that gaze behavior had no effect on avoidance in an orthogonal collision-avoidance task [41]. Murakami et al. reported that gaze information when passing one another is not used mutually with an oncoming passenger but is used individually to determine the direction to avoid [56]. These studies imply that body orientation, rather than partial body movements, is a more critical factor influencing avoidance behavior. Ratsamee et al. developed a collision-avoidance model incorporating human posture, facial orientation, and personal space, which allowed participants to better understand the robot's behavior and reported smoother avoidance [57].

## Experiment 1: AMR collision avoidance and body movement tracking

### Methods

**Participants.** Sixteen students from Toyohashi University of Technology participated in this experiment (see details in the Appendix).

**Apparatus. Environment:** We used a custom-made wheeled platform (Mega Rover Ver 2.1, Vstone) as the AMR (**Fig 1a**; see details in Appendix). To measure participants' motions, they wore five motion trackers (VIVE Tracker 3.0, HTC) on their waists, wrists, and ankles. The AMR was equipped with one tracker on its front (**Figs 1a** and **1b**). **Fig 1b** illustrates the experimental environment. The goal position was located 5 m in a straight line from the starting position of the participants. The theoretical points of collision (TPC) with the AMR were set at the center of the experimental environment, 3 m from the starting point. The AMR's initial positions were 1.5 m from the TPC, with five different angles (–40°, –20°, 0°, 20°, and 40°). The experimental field contained only one participant at a time, and only one AMR was used. The experimenter operated the control and tracking systems from outside the field.

**Procedure and task.** Initially, the AMR traveled randomly to one of the five starting positions. A buzzer on the AMR signaled the commencement of each trial. Following this cue, the participant began walking from the starting point to the goal at a comfortable pace. Concurrently, the AMR moved straight toward the TPC at a constant speed of 0.5 m/s. This speed was determined based on the outcomes of a preliminary experiment to ensure that participants did not feel intimidated by the robot. A trial concluded once the AMR had covered a distance of 3 m from its starting position. After this, both the participant and the AMR returned to their initial positions.

Participants were instructed to move either right or left to avoid the AMR. Specific guidelines were provided for collision avoidance: 1) avoid extreme evasive actions that could lead to an inefficient route to the goal, 2) do not make abrupt course changes, and 3) maintain a comfortable speed similar to what they would use in real-life situations. These instructions aimed to mimic a natural gait in a corridor or walkway, where collisions with an AMR would be unlikely. Before the experiment, participants were briefed on 1) the potential for the AMR to approach from any of its starting positions, 2) the

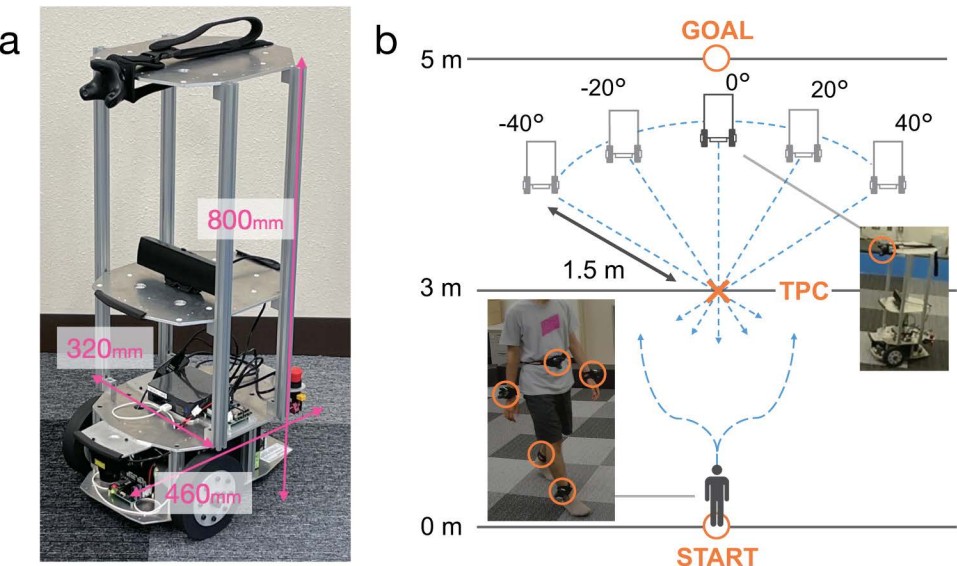

**Fig 1. Experimental setup.** (a) Autonomous mobile robot (AMR) in experiment. (b) Illustration of environment. Participant moves from starting point toward goal. Ratio of distances used in this panel differs from actual ratio used in study.

sequence of a trial, and 3) the total number of trials. No additional instructions that could influence their behavior were given. Each participant completed 50 trials (ten trials × five angles) in a random order.

**Data analysis.** For the analysis, we defined the start of an interval as the moment when the AMR emitted an alarm and the end as the moment when the participant had walked 5 m in a straight line or as close as possible to the goal point. Trials exceeding 10 s were excluded as artifacts, accounting for 1.5% of all trials. Additionally, two trials were excluded due to anomalous readings from the trackers caused by hardware issues. The average trial duration for participants was 7.07 ± 0.20 s. Trial times were normalized to a percentage value (0%–100%).

In each trial, we estimated the walking trajectory using the three-dimensional coordinates of the participants' heads, as recorded by the tracker on their waists. We then computed the probability of rightward avoidance based on the direction that showed the greatest change in mediolateral (ML) displacement at the TPC—the position reached by the participants after walking straight for 3 m. The average walking speed was calculated from the distance and time within the interval where participants walked between 1 and 3 m in the anteroposterior (AP) direction, excluding acceleration periods [39].

The Palamedes toolbox was utilized to fit a psychometric function for the probabilities of rightward avoidance at five angles, with the threshold and slope as free parameters [58,59]. In cases where the probabilities formed a step function, we fitted the data with only the threshold as the free parameter (this applied to four participants). One participant was excluded from the analysis due to an opposing trend that affected the psychometric function's performance. Thus, data from the remaining 15 participants were used for further analysis. To quantitatively assess the participants' bias toward an ML direction, we computed the starting angle at which each participant showed no consistent preference for avoiding the AMR on either the right or left side. This index reflects the participant's lateral avoidance tendency.

For motion analyses, we compared the following indices using the waist tracker: 1) rotation angle and 2) ML displacement. We first calculated changes in these indices for each avoidance direction over time. We quantitatively compared the onset times of deviations from a straight line to either side. Specifically, onset times were defined as the moments when the waist twisted more than 13.6° [60] and underwent an ML displacement greater than 0.25 m [39,61]. For further details, please refer to *Motion Analysis* in the **Results** section.

## Results

**Human collision-avoidance behavior.** **Fig 2** illustrates two profiles: 1) the average walking velocity and 2) the average distance between the participant and the AMR, based on each participant's data. Participants increased their pace during the first 20% of the trial and then maintained a steady walking speed of approximately 1.0 m/s between 30% and 80% of the trial duration (**Fig 2a**). Additionally, the average distance between participants and the AMR was reduced by 62% (**Fig 2b**).

**Fig 3a** shows the average walking trajectory for each avoidance direction. Participants successfully evaded the AMR in all trials, navigating symmetrically and maintaining distance from the TPC. Specifically, participants began to evade approximately 1.5 m in the AP direction after an average of 4.38 ± 0.27 s. The average walking speed was 1.03 ± 0.07 m/s, and the minimum distance to the AMR was 0.66 ± 0.18 m.

**Fig 3b** depicts the distribution of the estimated starting angle, showing the directions in which participants moved relative to the fitted individual curves. The average estimated starting angle was –3.47 ± 14.43° (vertical red line in **Fig 3b**), with no significant difference from the origin ($t(14) = -0.93$, $p = 0.37$, CI = [–11.46, 4.52]). Notably, the nine participants who evaded to the left had smaller absolute values and variances of their estimated starting angles compared to the six participants who evaded to the right. These findings suggest that participants did not exhibit a preference for a particular ML direction; rather, they chose avoidance directions without a common pattern.

A similar "no-bias" trend was observed when the AMR approached participants head-on (0°; green circle in **Fig 3c**), with an average probability of rightward evasion of 0.43 ± 0.28. However, this probability varied significantly among

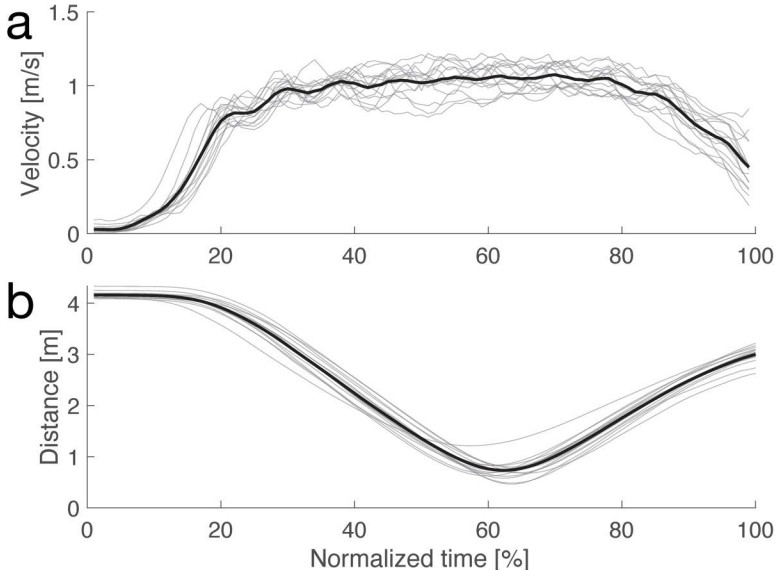

**Fig 2. Walking velocity and distance to AMR.** (a) Average velocity profile (thick) and that for each participant (thin). Horizontal axis represents normalized time between beginning and end of trials. Vertical axis represents velocity. (b) Average distance to AMR. Other formats are same as those for (a).

participants (**Fig 3d**), indicating that predicting the precise avoidance direction based solely on positional data is challenging when humans and AMRs cross paths.

**Motion analysis.** We sought to identify indicators for predicting the avoidance direction based on human behavior. Accurate prediction of this direction, if encoded in AMRs, could enhance the safety, smoothness, and comfort of human-robot interactions. To this end, we propose a method for predicting the avoidance direction based on the positions and rotations measured by motion trackers. We hypothesized that humans twist their waists in the preferred direction immediately before initiating an avoidance maneuver. Consequently, we focused on the waist angle as a key indicator, which can be measured using common sensors.

**Fig 4a** shows that the participants' average rotation angle (top) peaked more rapidly than the ML displacement (bottom). This suggests that the waist twists toward the target direction before the rest of the body moves to evade an approaching object. **Figs 4b** and **4c** demonstrate that the participants' waists twisted significantly before their entire bodies moved (left: $t(14) = 4.00$, $p < 0.005$, CI = [4.29, 14.21], $d = 1.03$; right: $t(14) = 3.53$, $p < 0.005$, CI = [1.67, 6.83], $d = 0.91$). These results indicate that, prior to movement, humans involuntarily twist their waists toward the avoidance direction.

## Experiment 2: AMR collision avoidance using motion trackers

The results of Experiment 1 suggest that waist rotation angle can predict human avoidance direction (**Fig 4**). Building on these findings, Experiment 2 introduced an avoidance algorithm for the AMR designed to anticipate human avoidance behavior. This experiment aimed to evaluate the effectiveness of the AMR's collision avoidance and the comfort levels associated with its tactics.

## Methods

### Participants

Nineteen students from Toyohashi University of Technology participated in this experiment (see details in Appendix).

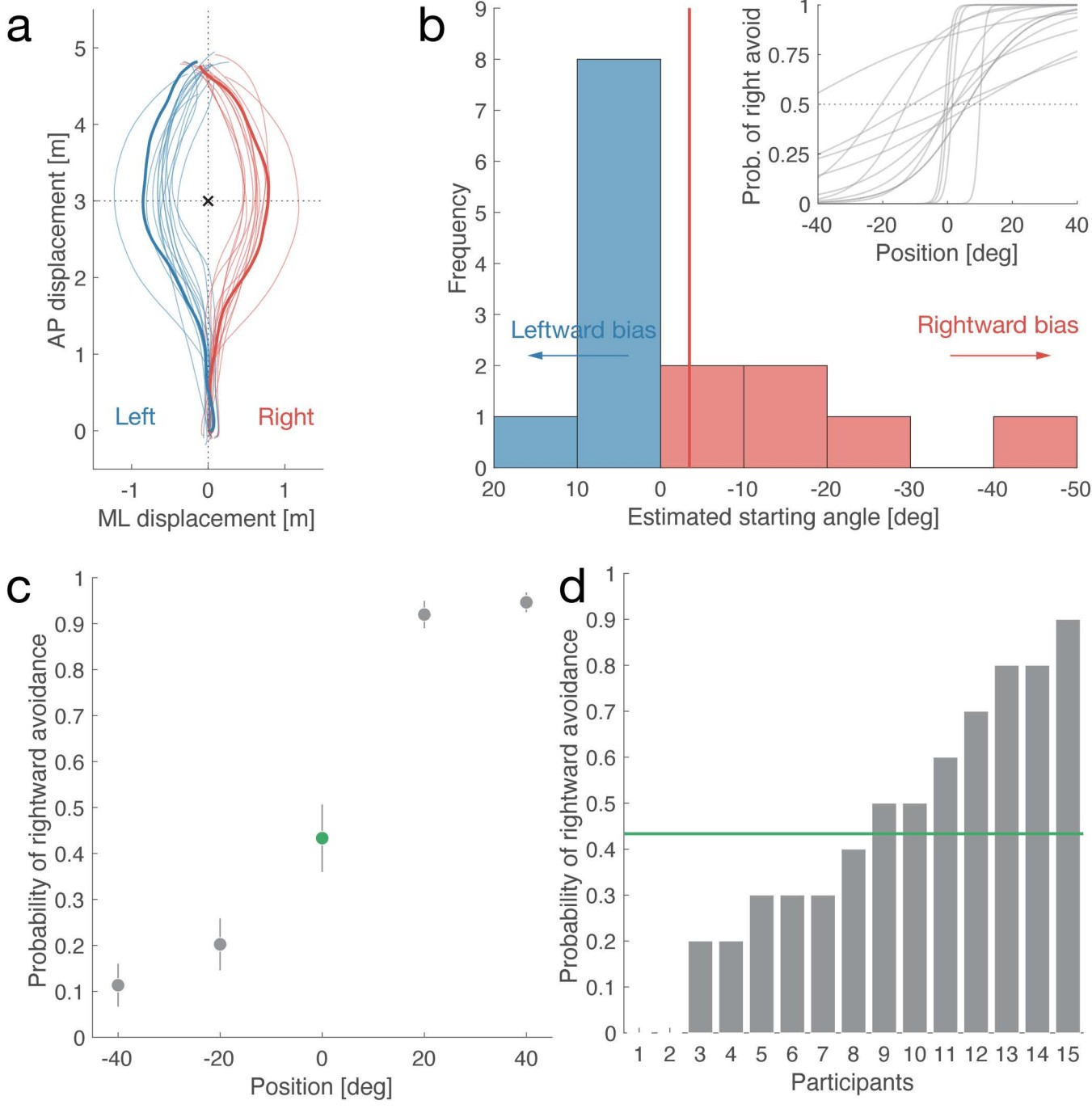

**Fig 3. Human collision-avoidance behavior.** (a) Walking trajectory. Horizontal and vertical axes represent mediolateral (ML) and anteroposterior (AP) displacements, respectively. Lines represent walking trajectories (red: rightward avoidance, blue: leftward avoidance) on average (thick) and for each participant (thin). Black cross represents location of theoretical point of collision (TPC). (b) Histogram of estimated starting angles at which participants showed no consistent side preference when avoiding the AMR. The horizontal axis represents the starting angle; negative values indicate rightward avoidance. The number of participants with rightward and leftward avoidance is six (red) and nine (blue), respectively. The red vertical line represents the average estimated starting angle (−3.47°). The image in the top-right corner illustrates the fitted individual psychometric curves used to estimate these angles. (c) Average probability of rightward avoidance (vertical axis) at initial AMR positions (horizontal axis). Error bars represent standard error of mean. (d) Probabilities of rightward avoidance for each participant (horizontal axis) in head-on direction. Vertical axis is same as in (c). Green horizontal line represents average (same as 0° in (c)).

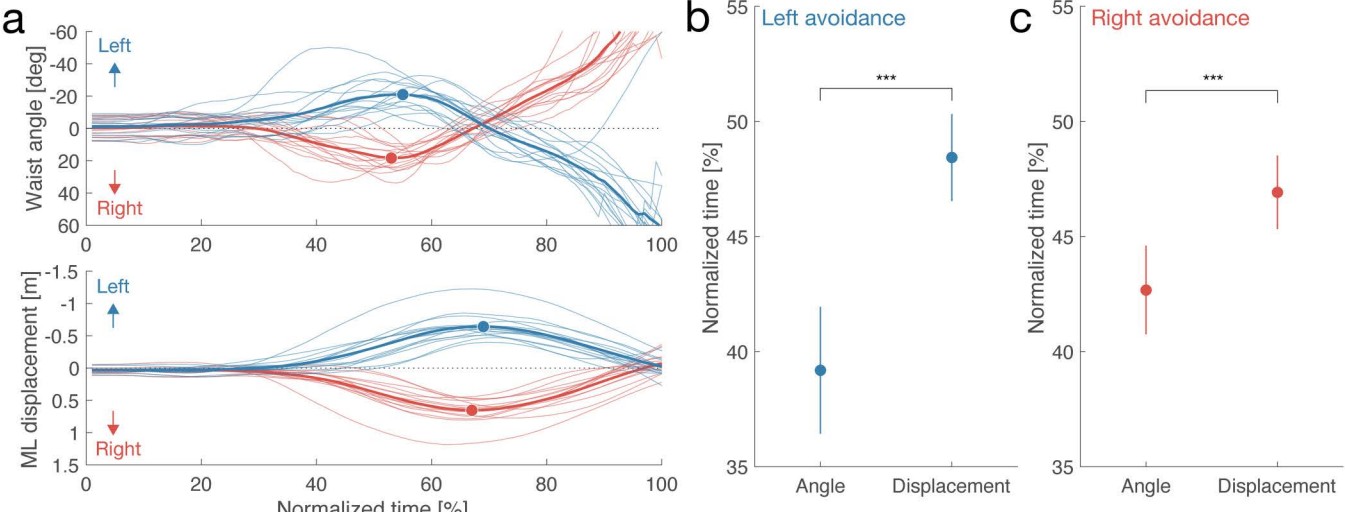

**Fig 4. Motion analysis.** (a) Changes in rotation angle and ML displacement at waist. Horizontal axis indicates normalized time between beginning and end of trials. Vertical axes indicate waist rotation angle (top) and ML displacement (bottom). Format is same as in **Fig 3a**. (b) Comparison of onset times between rotation angle and ML displacement in leftward avoidance case. Error bars represent standard error of mean. (c) This panel is same as (b) but for different avoidance direction. Asterisks indicate statistically significant differences in *t*-test; *** $p < 0.005$.

## Apparatus

**Control system for AMR:** The same AMR used in Experiment 1 was employed. For this experiment, the AMR was equipped with avoidance movement algorithms that used the participant's waist rotation angle to determine the direction to avoid and circumvent (see **Fig 5a**). The AMR initiated avoidance maneuvers at six different timings: Beginning, Early, Same, Late, Straight, and Stop (see **Fig 5b**). In the Same condition, the AMR began to avoid when the participant's waist rotation angle ($\theta$) reached their average waist rotation angle ($\theta_p$), as defined during the calibration phase. In contrast, in the Beginning condition, the AMR started to deflect to the left or right simultaneously with the start of the trial, ensuring immediate avoidance. In the Early and Late conditions, the AMR initiated deflection when the participant's waist rotation angle reached 0.5 and 1.5 times the average range of waist rotation angles, respectively. In the control conditions, the AMR continued moving forward in the Straight condition and stopped when the participant's waist rotation angle reached the average range in the Stop condition.

**Tracking system:** The tracking system used was the same as in Experiment 1.

**Environment:** The experimental environment remained consistent with Experiment 1; however, in this experiment, the AMR's initial position was always set 1.5 m from the TPC.

**Questionnaire:** A questionnaire assessed the timing of the AMR's avoidance movements and participants' comfort during each trial. It was administered via Google Sheets on a laptop and included two questions (**Fig 5c**). Question 1 (Q1) asked, "How much slower did the avoidance movement timing of the AMR seem relative to your own timing?" Responses were rated on a 5-point scale (1: very fast, 2: fast, 3: moderate, 4: slow, 5: very slow), with an additional option (6) for participants who felt the AMR did not execute any avoidance movement. Question 2 (Q2) asked, "How comfortable were you with walking in this trial?" Responses were rated on a 7-point scale (1: very uncomfortable, 2: uncomfortable, 3: slightly uncomfortable, 4: neutral, 5: slightly comfortable, 6: comfortable, 7: very comfortable).

## Procedure and task

In the calibration phase preceding the main experiment, participants' waist rotation angles were measured while they walked straight in the same field used for the subsequent experiments. Initially, participants positioned themselves at the

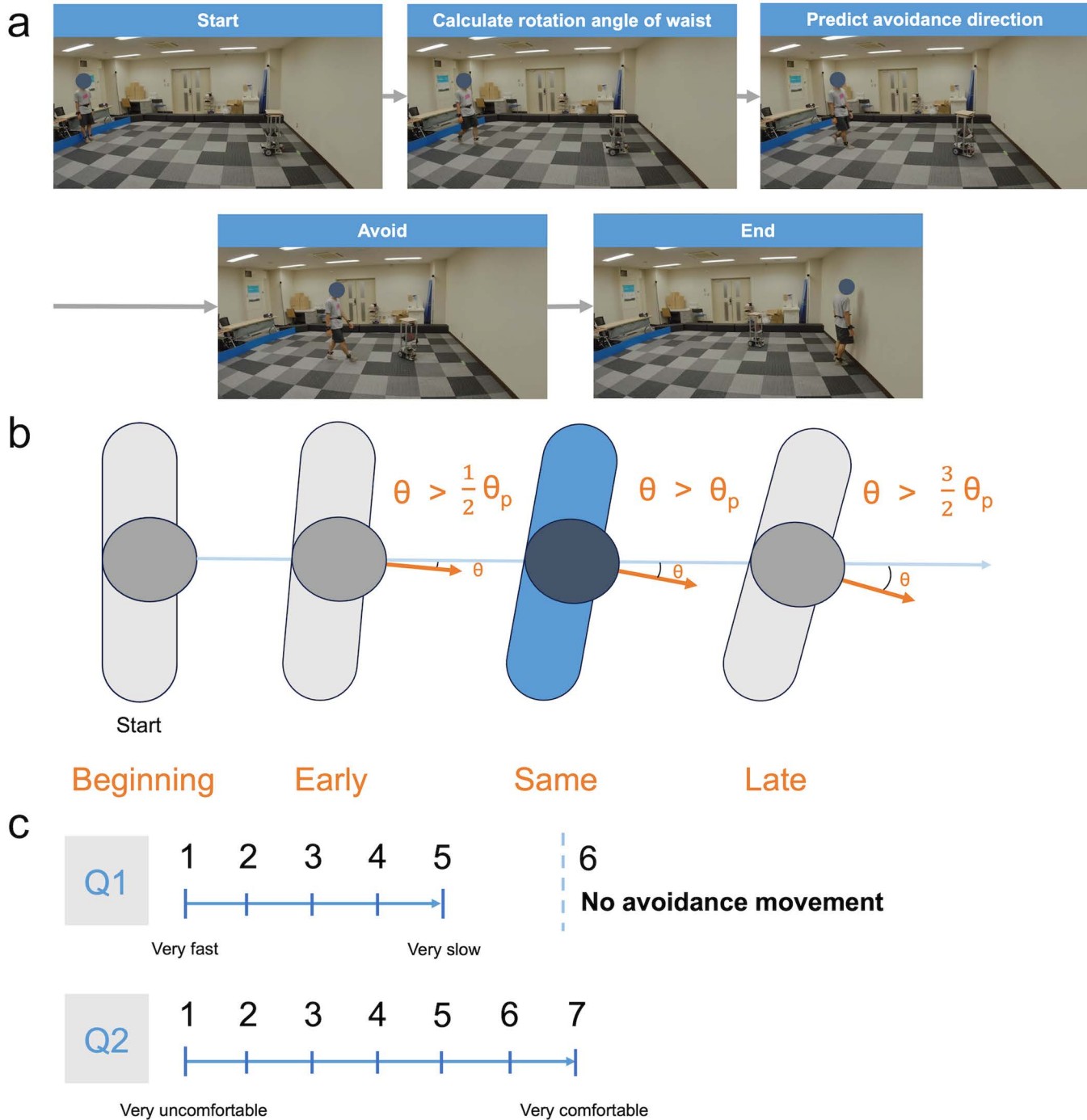

**Fig 5. Methods of Experiment 2.** (a) Illustration of predicting avoidance direction. (b) Different timings to detect waist rotation angle. (c) Two question-naires after each trial.

starting point. Upon receiving an auditory cue from a speaker, they walked at a natural, comfortable pace toward the goal and remained there until a second cue signaled the end of the trial. Each participant completed five trials.

In the primary phase, both participants and the AMR were positioned in their initial spots. A buzzer in the AMR signaled the start of the trial. Participants began walking from the starting point to the goal at a comfortable pace. As each participant took a step, the AMR moved straight toward the TPC at a constant speed of 0.5 m/s. The AMR executed six different avoidance movements based on the participant's waist angle, varying in the timing of the avoidance. At the end of the trial, the AMR emitted a buzzer to signal the trial's conclusion, and participants waited at the goal position until they heard the buzzer. After each trial, participants completed a questionnaire assessing the timing of the AMR's avoidance movements and their comfort while walking. Following this, both the participant and the AMR returned to their initial positions.

Participants were instructed to move from the start position to the goal position while avoiding a collision with the AMR, using the same techniques as in Experiment 1. Prior to the experiment, participants were informed about 1) the different timings of the AMR's avoidance movements, 2) the sequence of the trials, and 3) the number of trials. No additional instructions were given that could affect their behavior. A total of 24 trials (four trials × six conditions) were conducted randomly for each participant.

## Data analysis

Analysis commenced when participants moved 0.5 m from the start, triggered by the auditory signal from the AMR, and concluded when they reached the goal, with an average trial time of 5.32 ± 0.10 s. Trial times were converted to a percentage scale (0%–100%). During the calibration phase, the waist rotation angle range was calculated by subtracting the minimum angle from the maximum angle recorded in a trial. The average of these ranges across trials represented each participant's waist rotation threshold, indicating the initiation of avoidance.

For the questionnaire analysis, we calculated the probability of no-avoidance responses for Question 1 (Q1) to verify that the experiment was performed correctly as a control condition. For Question 2 (Q2), a one-way ANOVA was conducted using R with the statistical tool anovakun version 4.8.5. In the multiple comparison tests, $p$-values were adjusted using Shaffer's modified sequentially rejective Bonferroni procedure.

## Results

### Calibration phase

The average range of waist rotation angle among participants was 16.9 ± 1.90°. This value closely matched the deviation range from the straight line observed in a previous report [60].

### Avoiding tasks and questionnaires

**Fig 6a** illustrates the evaluations of avoidance timing for each condition. The timing gradually increased from the Beginning to the Late conditions, indicating that physical and subjective ratings were aligned. This confirms that the experiment was conducted correctly. The Straight condition was rated as the slowest, and the Stop condition was nearly identical to the Same condition, suggesting that participants accurately assessed the AMR's movement.

In contrast, when asked whether the AMR did not make avoidance movements (as shown in the plots in **Fig 6a**), 89% of responses reported that no-avoidance movements were observed in the Straight condition (0.89 ± 0.19). Additionally, 61% of responses indicated that no-avoidance movements were observed in the Stop condition (0.61 ± 0.44). These findings suggest that moving straight and stopping are less likely to be recognized as avoidance movements.

**Fig 6b** displays the average comfort levels associated with walking in each condition. Analysis of Question 2 revealed a significant main effect on comfort ($F(5,90) = 39.75$, $p < .001$, partial eta squared = 0.688). The Beginning condition had the highest comfort rating (6.18 ± 0.87), with significant differences observed between it and all other

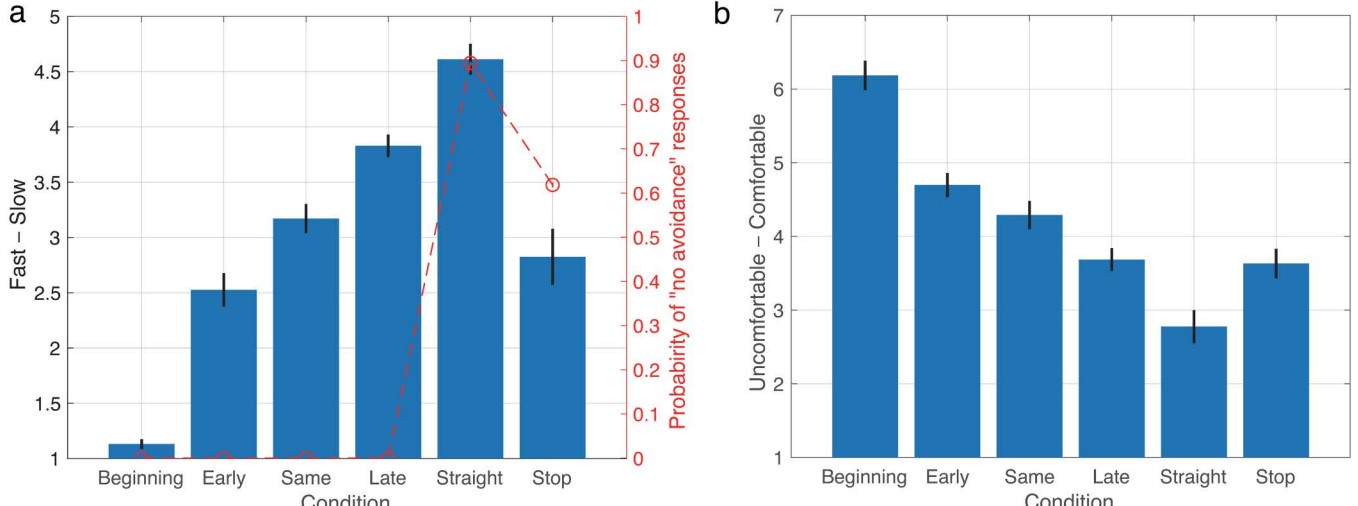

**Fig 6. Results of questionnaires.** (a) Average ratings of timing for avoidance among each condition. Horizontal axis indicates each condition. Left vertical axis indicates speed from very fast to very slow. Right vertical axis indicates probability of "no avoidance" responses. Error bars represent standard error of mean. (b) Average ratings of comfort for trial. Format was same as (a) except left vertical axis indicates comfort level from very uncomfortable to very comfortable.

conditions (Beginning > Early, Same, Late, Straight, Stop; all adjusted $p < .001$). The Early condition, with a comfort rating of $4.70 \pm 0.71$, was the second highest, showing significant differences from the Late, Straight, and Stop conditions (Early > Late, adjusted $p < .005$; Early > Straight, adjusted $p < .001$; Early > Stop, adjusted $p < .01$), but not from the Same condition. This indicates that participants preferred the AMR to avoid them earlier rather than later or not at all. Conversely, the Straight condition had the lowest comfort rating ($2.78 \pm 0.98$) and was rated significantly lower than all other conditions (Straight < Beginning, Early, Same, adjusted $p < .001$; Straight < Late and Stop, adjusted $p < .01$), suggesting discomfort when the AMR did not perform avoidance movements. The Stop condition, with a comfort rating of $3.63 \pm 0.88$, was nearly the same as the Late condition.

## Experiment 3: AMR collision avoidance using RGB camera with pose estimation

Experiment 2 demonstrated that participants felt more comfortable when the AMR avoided them earlier. However, predicting the direction of avoidance based on waist rotation angles relied entirely on motion trackers, limiting the method's effectiveness to indoor environments. To apply our avoidance algorithm in real-world settings, an alternative method for obtaining waist rotation data is necessary. Therefore, in Experiment 3, we introduced an RGB camera to capture human posture, including waist rotation, and compared the results with those from Experiment 2, using both the tracker and camera methods.

## Methods

### Participants

Twenty students from Toyohashi University of Technology participated in this experiment (see details in Appendix).

### Apparatus

**Pose estimation:** We utilized MediaPipe Pose for measuring human waist rotation. This lightweight tool requires no specific hardware and is compatible with various platforms [62]. It is suitable for our AMR due to the absence of a GPU for

real-time pose estimation. MediaPipe Pose tracks 33 body parts as landmarks, with each representing a point in 3D space using x, y, and z coordinates. This capability allows us to obtain precise joint angles, including the waist rotation angle, estimated from the coordinates of the hip landmark. Previous studies have reported that MediaPipe Pose performs comparably to other pose estimation libraries in video datasets [63], indicating sufficient accuracy for estimating the posture of individuals in motion.

Before conducting Experiment 3, we assessed whether MediaPipe Pose could effectively replace a motion tracker. To test this, we measured waist rotation using both the motion tracker and the camera simultaneously. The participant wore a motion tracker on their waist and rotated in front of the camera. **Fig 7b** illustrates the comparison between the waist rotation angles obtained from the tracker and the camera. The angles measured by MediaPipe Pose closely matched those from the motion tracker, suggesting that MediaPipe Pose can effectively substitute for the motion tracker.

**Control system for AMR:** We used a custom wheeled platform (Mega Rover Ver 3.0, Vstone) for the AMR (**Fig 7a**; see details in Appendix). The AMR was equipped with avoidance movement algorithms, similar to those used in Experiment 2, and included a method for obtaining human waist orientation using pose estimation by MediaPipe Pose. To facilitate pose estimation, the AMR was equipped with a RealSense Intel Depth Camera D435i.

**Environment:** The experimental environment was the same as in Experiments 1 and 2.

**Questionnaire:** The questionnaire was the same as in Experiment 2.

## Procedure and task

Before the main experiment, the calibration phase was conducted as previously described. This experiment comprised two sessions: one using a motion tracker, similar to Experiment 2, and the other employing pose estimation via a camera instead of the tracker. The order of the sessions was counterbalanced across participants. Each trial followed the same procedure as in Experiment 2, regardless of the session.

a

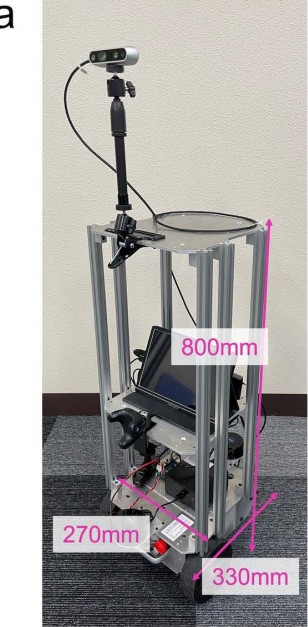

b

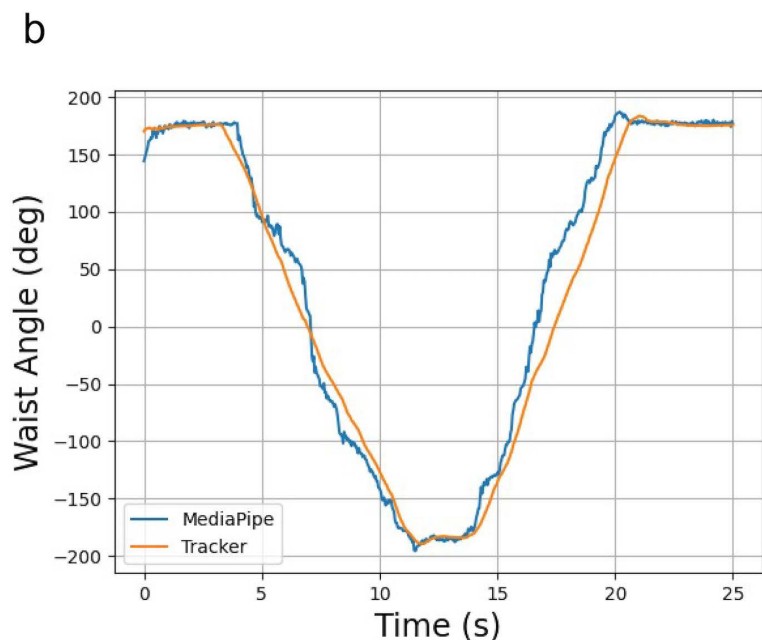

**Fig 7. Setup for Experiment 3.** (a) Autonomous mobile robot (AMR) in Experiment 3. (b) Comparison of waist rotation angles obtained from tracker and camera. Horizontal axis indicates time (seconds), while left vertical axis indicates waist rotation angle (degrees). Orange line represents data from motion tracker, and blue line represents data from MediaPipe Pose.

Participants received the same instructions and were allowed to take optional breaks between sessions. Each session consisted of 24 randomized trials (four trials × six conditions), resulting in a total of 48 trials for each participant.

## Data analysis

The analysis procedures for the beginning and end of trials were consistent with those used in Experiment 2. The average trial time was $7.70 \pm 0.12$ s for the tracker session and $7.88 \pm 0.15$ s for the camera session. Four trials were excluded from the average trial time calculation due to missing end-of-trial cues caused by hardware issues. However, since no other issues were present apart from the missing cues, the data from these trials were included in the subsequent analysis. The questionnaire data were analyzed using the same methods as in Experiment 2.

## Results

**Calibration phase.** The average range of waist rotation angles among participants was $14.9 \pm 2.80°$, similar to the results observed in Experiment 2 and consistent with the previous study [60].

**Avoiding tasks and questionnaires.** No collisions occurred between participants and the AMR. However, predictions of the AMR's avoidance direction occasionally failed under certain conditions (Early, Same, Late). Specifically, the average probability of failed predictions was 4.39% in the tracker session and 15.4% in the camera session.

Figs 8a and 8c show evaluations of the timing for avoidance under each condition for the tracker and camera sessions, respectively. In the tracker session, evaluation scores gradually increased from the Beginning to the Late conditions, reflecting a pattern consistent with the results of Experiment 2 (Fig 6a). This indicates that updating the software and changing the hardware did not negatively impact the experimental procedure. In the camera session, evaluation scores slightly increased from the Early to the Late conditions compared to the tracker session. Despite this, the overall tendencies in both sessions were similar, confirming that the experiment was conducted correctly in both cases.

Conversely, when participants were asked whether the AMR made avoidance movements (as shown in Figs 8a and 8c), 99% of responses indicated that avoidance movements were not observed in the Straight condition ($0.99 \pm 0.0030$), and 61% indicated that avoidance movements were not observed in the Stop condition ($0.61 \pm 0.025$) during the tracker session.

In the camera session, 95% of responses indicated no-avoidance movements in the Straight condition ($0.95 \pm 0.0070$), while 67% indicated no-avoidance movements in the Stop condition ($0.67 \pm 0.021$). Notably, the value for the Late condition was higher in the camera session compared to the tracker session. This discrepancy may be attributed to variations in the effectiveness of pose estimation, which could impact the AMR's ability to execute avoidance actions.

Figs 8b and 8d show the average comfort of walking in each session. In the tracker session, there was a significant main effect on comfort ($F(5, 90) = 35.33$, $p < .001$, partial eta squared $= 0.663$). The comfort level in the Beginning condition was $6.24 \pm 0.22$, which was significantly higher than that of the Same, Late, Straight, and Stop conditions (all adjusted $p < .001$). The Early condition, with a comfort level of $5.58 \pm 0.20$, also showed significant differences compared to the Same, Late, Straight, and Stop conditions (all adjusted $p < .001$). In contrast, the Straight condition had the lowest comfort level at $3.22 \pm 0.22$. There were no significant differences among the Same, Late, and Stop conditions, unlike the results from Experiment 2. These findings suggest that participants preferred the AMR to avoid them earlier rather than at the same time, later, or not at all.

In the camera session, there was a significant main effect on comfort as well ($F(5,90) = 24.13$, $p < .001$, partial eta squared $= 0.573$). The comfort level in the Beginning condition was $6.30 \pm 0.27$, and it was significantly higher than that in the Early, Same, Late, Straight, and Stop conditions (all adjusted $p < .001$). The Early condition had a comfort level of $4.99 \pm 0.34$, with significant differences observed compared to the Same, Late, and Stop conditions ($p < .05$; Early > Straight, adjusted $p < .001$). In contrast, the Straight condition had the lowest comfort level at $3.09 \pm 0.22$, which was significantly lower than the Beginning and Early conditions (adjusted $p < .001$ for Straight < Beginning and Early; adjusted

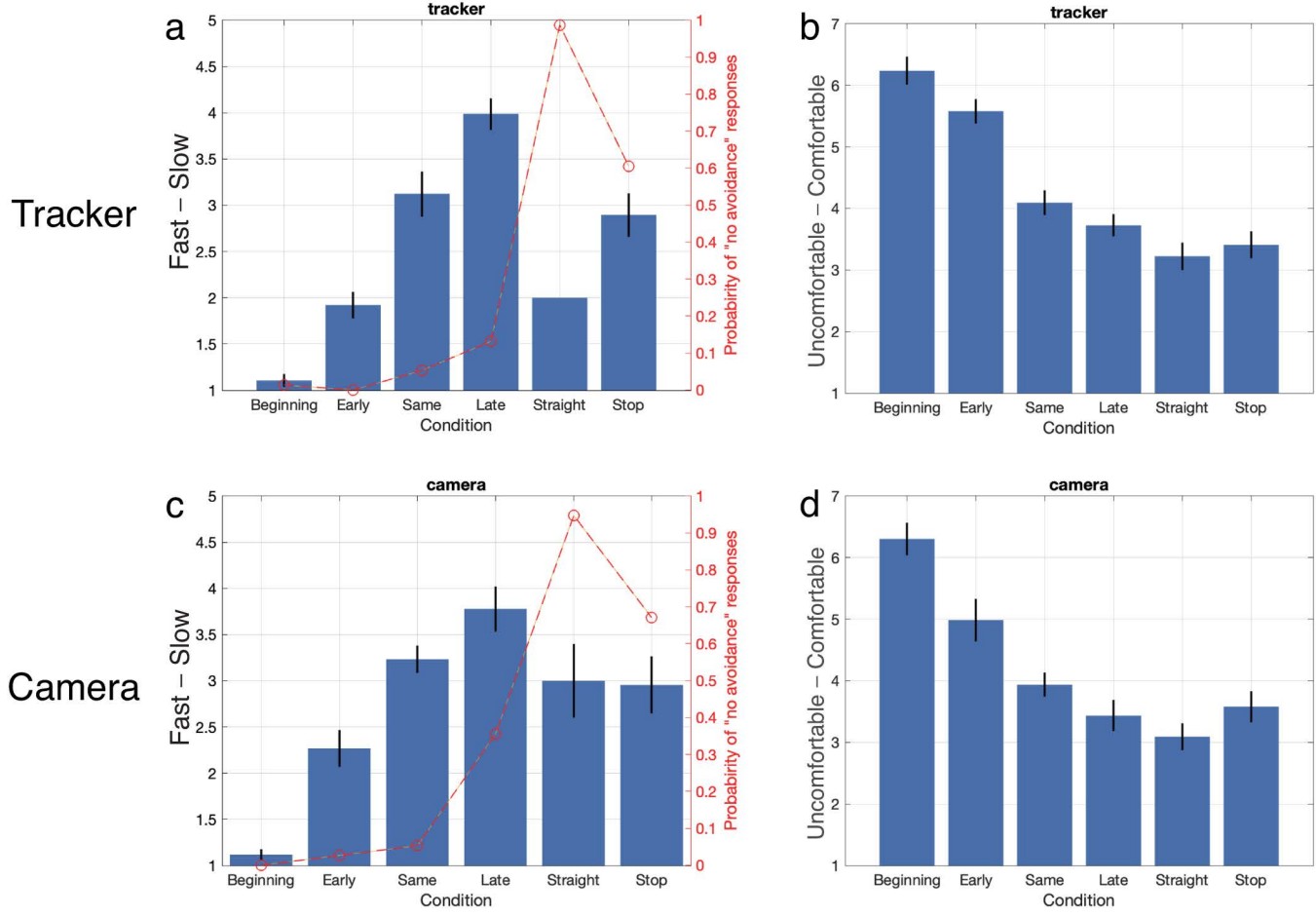

**Fig 8. Results of Experiment 3.** (a) Average ratings of timing for avoidance in each condition during tracker session. Horizontal axis indicates each condition. Left vertical axis indicates speed from very fast to very slow. Right vertical axis indicates probability of "no avoidance" responses. Error bars represent standard error of mean. (b) Average ratings of comfort for trial in tracker session. Horizontal axis indicates each condition. Left vertical axis indicates level of comfort from very uncomfortable to very comfortable. (c) Average ratings of timing for avoidance in each condition during camera session. (d) Average ratings of comfort for trial in camera session.

$p < .05$ for Straight < Same). The comfort levels in the camera session followed a similar trend to those in the tracker session, and there was a strong correlation between the two sessions ($r = .98$, $p < .001$). This suggests that pose estimation could be an effective substitute for a motion tracker. However, unlike the tracker session, the Beginning condition in the camera session was significantly higher than the Early condition. Additionally, the comfort level in the Early condition was lower in the camera session compared to the tracker session.

## Discussion

### General summary

In this study, we investigated human collision-avoidance behavior in response to an AMR approaching head-on, as explored in Experiment 1. We found that the waist rotation angle served as a useful cue for predicting avoidance direction. Experiment 2 demonstrated that the AMR could effectively avoid humans based on this waist rotation angle, providing greater comfort compared to conditions where the AMR either went straight or stopped. In Experiment 3, we introduced pose estimation

as an alternative to motion tracking for detecting waist rotation angles. The results demonstrated a similar pattern to those observed in Experiment 2. Specifically, earlier avoidance by the AMR was preferred over later avoidance and no-avoidance strategies. However, the pose estimation method exhibited a higher probability of failed predictions compared to the motion tracker. Despite this, a strong correlation existed between the two methods regarding walking comfort. This finding supports the potential of using waist rotation angles to predict avoidance directions in real-world scenarios, provided that pose estimation accuracy is improved. Neggers et al. reported that a robot's predictable movements enable people to anticipate its intended direction, thereby influencing their comfort level during interactions [47]. In our study, early avoidance by the AMR was perceived as a predictable action, contributing to higher comfort levels. These results underscore the effectiveness of using waist rotation angles as indicators for predicting human avoidance directions and highlight the importance of communicating avoidance actions early when an AMR and a human are approaching head-on.

## Discussion of experiment 1

Experiment 1 examined human strategies for evading an AMR approaching from various directions by recording participants' movements. As illustrated in **Fig 3a**, participants' walking trajectories followed trends consistent with those reported in the existing literature [39,40]. This suggests that human avoidance trajectories are robust, regardless of the attributes of the target. However, the average walking speed in our study was marginally lower (1.03 m/s) than that reported in previous studies, indicating that participants may have exercised caution to minimize collision risks [64]. We did not collect reference data on participants' comfortable speeds without the AMR present. Additionally, the minimum distance maintained from the AMR was shorter (0.66 m). One possible explanation is that participants were influenced by the AMR's smaller size compared to their own [30].

No significant bias was observed in either direction, although the estimated starting angle varied among participants based on ease of evasion (**Fig 3b**). This result may reflect that the right/left turn preference of 40% of the participants remains unchanged [65]. Souza Silva et al. suggested that a rightward avoidance bias might be influenced by regional driving habits [39]. In contrast, our results showed no significant bias, with a slight preference for leftward movement (rightward direction probability: 43%), possibly influenced by similar factors. Thus, this finding may not be universally applicable, as various factors, including cultural norms, can affect evasion strategies.

Our findings are consistent with previous studies on human evasion of cylindrical objects in virtual environments [39]. This comparison suggests that collision-avoidance strategies may vary based on the target object [48], implying that humans adapt their avoidance strategies to different targets, such as AMRs. More importantly, these results highlight the complexity of predicting human avoidance direction when an AMR approaches head-on.

Participants demonstrated a tendency to twist their waists before lateral displacement (**Figs 4a** and **4b**), suggesting that humans involuntarily rotate their waists in the direction of evasion. Previous research has shown that changes in the center of mass can indicate different coordination strategies between steering and circumventing obstacles [27]. Additionally, involuntary waist rotations, such as those induced by the hanger reflex, can alter walking direction [66,67]. These observations underscore the importance of focusing on waist angle as a crucial indicator in determining walking direction, regardless of intention or spontaneity.

## Discussion of experiment 2

Experiment 2 investigated how participants avoided an AMR by predicting the direction of their avoidance and assessing their responses regarding speed and comfort level (Q1 and Q2; **Fig 5c**) during the trial. The analysis of comfort levels across different conditions (**Fig 6b**) revealed that participants felt more comfortable when the AMR avoided them sooner. This result may reflect how participants' walking experience varies depending on whether they can proceed directly to their goal or must detour. Specifically, participants walked directly to the goal in the Beginning condition, while detouring in other conditions required additional planning, likely reducing their comfort. Neggers et al. observed

a small negative correlation between walking deviation and comfort [47]. Furthermore, comfort levels in the Early and Same conditions were not as high as in the Beginning condition, possibly due to the AMR's avoidance movements. Previous research has demonstrated that smoother and safer interactions between humans and robots occur when social cues—such as body posture, facial orientation, and personal space—are considered [57]. These social cues are utilized in everyday human interactions to predict others' movements, such as interpreting gaze direction [52]. The lack of such social cues in the AMR's avoidance strategy likely contributed to the discomfort observed.

Additionally, our results showed that the Straight condition resulted in lower comfort compared to other avoidance conditions. This suggests that the AMR's behaviors, such as avoiding or stopping, alleviate some discomfort associated with potential collisions. The Stop condition yielded similar or slightly lower comfort levels, indicating that AMRs anticipating human movement to avoid collisions may be perceived as more effective and safer. For instance, AMRs used in consumer settings, such as dish delivery in restaurants or office cleaning, generally stop to avoid collisions. Our findings suggest that while these behaviors are effective, there still exists potential for improvement in their implementation.

### Discussion of experiment 3

Experiment 3 investigated whether pose estimation could substitute for a motion tracker to measure waist rotation, using the same procedure as Experiment 2. Consistent with Experiment 2, comfort levels were higher when the AMR avoided participants quickly and lowest when it did not avoid participants in the Straight condition, for both the tracker and camera sections (Q2; Figs 8b and 8d). A strong correlation was observed between the tracker and camera sections. However, the average comfort level in the Early condition was lower in the camera section compared to the tracker section. This discrepancy may be attributed to lower evaluations of avoidance timing in the Early condition with the camera, potentially due to the slower processing speed of pose estimation. While the motion tracker operated at 90 Hz, the pose estimation data was updated at a rate below 90 Hz.

Furthermore, unlike in Experiment 2, there were no significant differences between the Straight and Late conditions in both sections. In the tracker section, no significant differences were observed even in the Same condition. This suggests that late or simultaneous avoidance movements do not contribute to comfort. In these cases, the slower evasion time of the AMR makes it challenging for participants to predict its avoidance direction. According to Carton et al., predictable movements positively influence others' locomotion plans within the same environment [43]. Difficulty in predicting AMR movements may, therefore, contribute to discomfort.

### Indicators from body movements

Although our focus was on waist movement to predict avoidance direction, more precise predictions could be achieved by analyzing other body movements. Studies have demonstrated that signals from various body parts, such as gaze direction, can predict avoidance behavior [52,68,69]. For instance, the gaze direction of a person approaching head-on influences their trajectory [52]. This suggests that humans use gaze direction as a cue for their movement intentions, even though head rotation is not necessarily related to walking direction [70]. Similarly, shoulder rotation is related to the passability of apertures [29]. Additionally, wrist and ankle movements may be crucial indicators for predicting avoidance direction. For example, motion and gait features are used to recognize Parkinson's disease [71]. Recent studies have also shown that the landing foot during walking can predict avoidance direction [50]. Furthermore, a previous study found no significant difference between the mean onset of ML deviation and trunk angle in human walkers crossing obstacles [27]. We hypothesize that body parts closer to the legs, such as the waist, are more directly associated with walking direction. Overall, our findings suggest that the direction of human avoidance can be predicted based on body orientation even before movement begins.

### Limitations

This study has several limitations. First, we did not account for the sequence in which participants and the AMR crossed paths, nor did we provide relevant instructions regarding this sequence. Silva et al. proposed an effective measure based

on robotics that categorized collision avoidance into "first" or "subsequent" interactions between humans and robots [20]. In contrast, our experiment did not explicitly instruct participants to evade either before or after the TPC. Second, while the AMR speed was consistently set to 0.5 m/s across both experiments, human responses varied with changes in AMR speed. Third, to better simulate real-world scenarios, participants should interact with the AMR after reaching a steady-state self-selected walking speed. This implies that a larger space or an environment with multiple objects would be more effective in bridging the gap between laboratory settings and real-world conditions. Fourth, people generally do not expect AMRs to avoid them based on their movements, which may alter their perceptions during the experiment. This expectation might be disrupted by incorporating unpredictable movements that the AMR avoids in the same direction as the participants. Finally, the current study was conducted with a single participant. For practical applications, future research should include multiple individuals to better reflect real-world scenarios.

## Conclusion

In this study, we investigated human avoidance behavior when encountering an AMR approaching head-on and identified the waist rotation angle as a key indicator for predicting the direction of avoidance. Our findings suggest that using this indicator can enhance the comfort of human-robot interactions, particularly when the AMR performs early avoidance actions. Additionally, we developed and tested a non-contact pose estimation method as an alternative to direct motion tracking, which typically requires participants to wear sensors. Although the pose estimation method demonstrated slightly lower accuracy compared to the motion tracker, it showed promise as a practical tool for real-world applications by eliminating the need for physical contact.

This study contributes to the understanding of human behavior in human-robot interactions by demonstrating the utility of waist rotation as a predictive cue. The successful implementation of a non-contact pose estimation method also expands the potential applications of AMRs, although further refinement is necessary to enhance its accuracy and robustness. The importance of early, predictable movements by AMRs was reinforced, as these movements significantly increase user comfort.

While these findings are promising, additional research is required to validate the robustness of the non-contact pose estimation method and explore other factors that may influence human comfort and safety in AMR interactions. Future studies should focus on improving the accuracy of non-contact detection methods and examining their applicability in a broader range of environments.

## Supporting information

**S1. Additional methodological details and Appendix text.**
(DOCX)

## Acknowledgments

We appreciate the support from all the institutions and individuals who contributed to this research.

## Author contributions

**Conceptualization:** Hideki Tamura.

**Data curation:** Tatsuto Yamauchi, Hideki Tamura.

**Formal analysis:** Tatsuto Yamauchi, Hideki Tamura.

**Funding acquisition:** Hideki Tamura, Tetsuto Minami, Shigeki Nakauchi.

**Investigation:** Tatsuto Yamauchi, Hideki Tamura.

**Methodology:** Tatsuto Yamauchi, Hideki Tamura.

 

**Project administration:** Hideki Tamura.

**Resources:** Hideki Tamura, Tetsuto Minami, Shigeki Nakauchi.

**Software:** Tatsuto Yamauchi, Hideki Tamura, Shigeki Nakauchi.

**Supervision:** Hideki Tamura, Tetsuto Minami, Shigeki Nakauchi.

**Validation:** Tatsuto Yamauchi, Hideki Tamura, Tetsuto Minami, Shigeki Nakauchi.

**Visualization:** Tatsuto Yamauchi, Hideki Tamura.

**Writing – original draft:** Tatsuto Yamauchi, Hideki Tamura.

**Writing – review & editing:** Tatsuto Yamauchi, Hideki Tamura, Tetsuto Minami, Shigeki Nakauchi.

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
