## [Decision Letter · Decision Letter 0]

29 Jan 2025

PONE-D-24-41357Waist rotation angle as an indicator of probable human collision avoidance direction for autonomous mobile robotsPLOS ONE

Dear Dr. Tamura,

Thank you for submitting your manuscript to PLOS ONE. After careful consideration, we feel that it has merit but does not fully meet PLOS ONE’s publication criteria as it currently stands. Therefore, we invite you to submit a revised version of the manuscript that addresses the points raised during the review process.

We look forward to receiving your revised manuscript.

Kind regards,

silas onyango awuor, msc

Academic Editor

PLOS ONE

Journal Requirements:

“This paper is based on the results from project JPNP20004, subsidized by the New Energy and Industrial Technology Development Organization (NEDO). This work was supported by the CASIO SCIENCE PROMOTION FOUNDATION (39-55) funds, the Foundation of Public Interest of Tatematsu, and the 2021 Toyohashi University of Technology President Funding (Young Researchers).”

“This paper is based on the results from project JPNP20004, subsidized by the New Energy and Industrial Technology Development Organization (NEDO). This work was supported by the CASIO SCIENCE PROMOTION FOUNDATION (39-55) funds, the Foundation of Public Interest of Tatematsu, and the 2021 Toyohashi University of Technology President Funding (Young Researchers).”

 “This paper is based on the results from project JPNP20004, subsidized by the New Energy and Industrial Technology Development Organization (NEDO). This work was supported by the CASIO SCIENCE PROMOTION FOUNDATION (39-55) funds, the Foundation of Public Interest of Tatematsu, and the 2021 Toyohashi University of Technology President Funding (Young Researchers).”

4. Please remove your figures from within your manuscript file, leaving only the individual TIFF/EPS image files, uploaded separately. These will be automatically included in the reviewers’ PDF

Additional Editor Comments (if provided):

Kindly act on the raised comments

Reviewers' comments:

Reviewer's Responses to Questions

**Comments to the Author**

1. Is the manuscript technically sound, and do the data support the conclusions?

Reviewer #1: Yes

2. Has the statistical analysis been performed appropriately and rigorously? 

Reviewer #1: Yes

3. Have the authors made all data underlying the findings in their manuscript fully available?

Reviewer #1: Yes

4. Is the manuscript presented in an intelligible fashion and written in standard English?

Reviewer #1: Yes

5. Review Comments to the Author

Reviewer #1: This paper presents a series of experiments to study the interaction between AMR and human participants. The authors conducted three experiments to develop a method for collision avoidance that would be well perceived by human participants. In the first experiment human participants had to walk against AMR coming from different angles to study which behavior of human participants can hint for future actions. In particular, the authors intended to study whether it is possible to predict the direction in which people would move to avoid the AMR. Results showed that waist rotation occurs slightly before the actual steering maneuver and can therefore be used as a predictor for avoidance direction. Based on these results the authors implemented a collision avoidance algorithm in the robot and tested it in experiments. In the first experiment data from motion tracker on human were used to guide the robot. However, since this would not possible in reality the authors later also tested another method using information gained from a sensor on the robot. Results show that participants prefer an early steering maneuver from the robot and judge it more “comfortable.”

The paper is interesting and well presented. All details needed for reproducing the findings are provided and literature review is generally complete and satisfactory. The structure follows a logical structure in which exploratory experiments are presented first and implementation based on the results follow to show how the authors intend to make their solution usable in reality.

However, I must admit I found the paper quite verbose. I understand that providing all details is important, but sometimes I found the amount of information overwhelming. For example (but it is not the only one), Fig. 6 contains all the value for the scale which is also reported in the text. I believe something like “from very fast to very slow” or “from very uncomfortable to very comfortable” would be enough. Also, much of the text in the discussion can be found in the presentation of the results. The fact that waist rotation can predict avoidance direction is repeated a large number of times in the manuscript and I am not sure whether it is really important to stress on something that is already clear when results are presented.

Since this already the second review of this manuscript and I do not see the verbosity as an obstacle for publication (the paper is good, so for me can be published with minimal modification), I do not want to force the authors into an extensive round of review. Nonetheless, I believe that making the paper more “compact” can help them making it readable to a larger audience. For this reason I want to give a chance to the authors to review their paper, but I also want to make it clear it is not a condition for acceptance.

As a personal suggestion I would propose to only keep the aspects related to the experiment in the main text and move all minor technical details to an appendix. For example, average age of the participants, date of the experiment, ethical approval, OS and model of the robot, CV tool used, etc. can be moved to an appendix. I think they are not really relevant to understand the experiments and are only needed in case someone want to reproduce it (so they must be kept, but probably not in the main text).

Some additional comments are given as follows:

1. Abstract: Readers cannot know that is your Experiment 2 and 3 so I suggest removing them. Also, since only experiments 2 and 3 are mentioned, one would wonder where it Experiment 1. Of course, that is clear in the manuscript, but abstract is read before the main text.

2. Fig. 5: I found if difficult to understand and generally not standalone. Maybe you can consider providing a number of snapshots, like in the old animation movies. Or use colors to show the change in time. For example, as robot and person move toward each other the color gets brighter (black, dark gray, light gray, etc.). b is also not that clear. The angle of the arrow changes a little, making it difficult to understand. What about using a top view where a person is moving the waist? In c I found “slower” and “more comfortable” redundant and creating more confusion.

3. Fig. 6: As already said, I found the reporting of the scale here redundant.

4. Fig. 7: I believe you are using MATLAB. If so, why don’t you use unwrap to “join” the angles and avoid the jump from 0 to 360? You can have negative values and the graphs looks clearer. The jump you see looks like there is some data loss or something strange happened, while in reality is simply related to the definition of the angle.

5. As already mentioned I do not want to force you into an additional work. I genuinely believe that the two papers below are related to your work, so maybe you can find them useful for this paper or in a follow-up study.

Jia, Xiaolu, et al. "Experimental study on the evading behavior of individual pedestrians when confronting with an obstacle in a corridor." Physica A: Statistical Mechanics and its Applications 531 (2019): 121735.

Murakami, Hisashi, et al. "Spontaneous behavioral coordination between avoiding pedestrians requires mutual anticipation rather than mutual gaze." Iscience 25.11 (2022).

6. PLOS authors have the option to publish the peer review history of their article (what does this mean? ). If published, this will include your full peer review and any attached files.

**Do you want your identity to be public for this peer review?** For information about this choice, including consent withdrawal, please see our Privacy Policy .

Reviewer #1: **Yes: ** Claudio Feliciani

---

## [Author Response · Author response to Decision Letter 1]

8 Feb 2025

Response Letter

We thank the editor and reviewers for their time and effort in reviewing our manuscript. The changes we have implemented based on their insightful comments, suggestions, and feedback have significantly improved our work. We have made revisions to strengthen the manuscript. Additionally, we utilized a professional English language editing service to improve the language quality of the manuscript. Detailed responses to the reviewers’ comments are provided below.

Reviewer #1: This paper presents a series of experiments to study the interaction between AMR and human participants. The authors conducted three experiments to develop a method for collision avoidance that would be well perceived by human participants. In the first experiment human participants had to walk against AMR coming from different angles to study which behavior of human participants can hint for future actions. In particular, the authors intended to study whether it is possible to predict the direction in which people would move to avoid the AMR. Results showed that waist rotation occurs slightly before the actual steering maneuver and can therefore be used as a predictor for avoidance direction. Based on these results the authors implemented a collision avoidance algorithm in the robot and tested it in experiments. In the first experiment data from motion tracker on human were used to guide the robot. However, since this would not possible in reality the authors later also tested another method using information gained from a sensor on the robot. Results show that participants prefer an early steering maneuver from the robot and judge it more “comfortable.”

Thank you for summarizing our manuscript.

The paper is interesting and well presented. All details needed for reproducing the findings are provided and literature review is generally complete and satisfactory. The structure follows a logical structure in which exploratory experiments are presented first and implementation based on the results follow to show how the authors intend to make their solution usable in reality.

Thank you very much for your positive feedback.

However, I must admit I found the paper quite verbose. I understand that providing all details is important, but sometimes I found the amount of information overwhelming. For example (but it is not the only one), Fig. 6 contains all the value for the scale which is also reported in the text. I believe something like “from very fast to very slow” or “from very uncomfortable to very comfortable” would be enough.

Thank you for your specific feedback. We have simplified the y-axis label and caption in Figs. 6 and 8.

Also, much of the text in the discussion can be found in the presentation of the results. The fact that waist rotation can predict avoidance direction is repeated a large number of times in the manuscript and I am not sure whether it is really important to stress on something that is already clear when results are presented.

Thank you for your feedback. We have revised the discussion to eliminate redundancy with the results section.

Since this already the second review of this manuscript and I do not see the verbosity as an obstacle for publication (the paper is good, so for me can be published with minimal modification), I do not want to force the authors into an extensive round of review. Nonetheless, I believe that making the paper more “compact” can help them making it readable to a larger audience. For this reason I want to give a chance to the authors to review their paper, but I also want to make it clear it is not a condition for acceptance.

Thank you for your comments. As you suggested, we have eliminated redundant sentences. Additionally, we used an English proofreading service to ensure the coherence and flow of the text remain intact.

As a personal suggestion I would propose to only keep the aspects related to the experiment in the main text and move all minor technical details to an appendix. For example, average age of the participants, date of the experiment, ethical approval, OS and model of the robot, CV tool used, etc. can be moved to an appendix. I think they are not really relevant to understand the experiments and are only needed in case someone want to reproduce it (so they must be kept, but probably not in the main text).

Thank you for your suggestion. As you mentioned, we have reduced the main text in the Methods section and moved minor technical details to the Appendix to make the paper more concise. Specifically, parts of the Participants and Apparatus sections have been relocated. However, for Figs. 5 and 7, which illustrate the methodology, we retained the relevant descriptions in the main text to maintain consistency with the figure explanations, minimizing the information moved to the Appendix.

Some additional comments are given as follows:

1. Abstract: Readers cannot know that is your Experiment 2 and 3 so I suggest removing them. Also, since only experiments 2 and 3 are mentioned, one would wonder where it Experiment 1. Of course, that is clear in the manuscript, but abstract is read before the main text.

Thank you for your feedback. As you suggested, we have removed explicit references to Experiment 2 and Experiment 3 from the abstract to improve readability.

2. Fig. 5: I found if difficult to understand and generally not standalone. Maybe you can consider providing a number of snapshots, like in the old animation movies. Or use colors to show the change in time. For example, as robot and person move toward each other the color gets brighter (black, dark gray, light gray, etc.). b is also not that clear. The angle of the arrow changes a little, making it difficult to understand. What about using a top view where a person is moving the waist? In c I found “slower” and “more comfortable” redundant and creating more confusion.

Thank you for your suggestion. We have revised Fig. 5 as follows:

Fig. 5a: As you suggested, we illustrated the progression from the beginning to the end of a trial using snapshots from a video.

Fig. 5b: We clarified the movement of the participants’ waist by using a schematic top-down view, showing the changes in the waist angle (θ) for each condition (Beginning, Early, Same, Late).

Fig. 5c: We simplified the labels to improve readability.

3. Fig. 6: As already said, I found the reporting of the scale here redundant.

As mentioned above, we have revised the y-axis label and caption in Figs. 6 and 8.

4. Fig. 7: I believe you are using MATLAB. If so, why don’t you use unwrap to “join” the angles and avoid the jump from 0 to 360? You can have negative values and the graphs looks clearer. The jump you see looks like there is some data loss or something strange happened, while in reality is simply related to the definition of the angle.

Thank you for your suggestion. We applied the unwrap function in MATLAB to ensure a continuous representation of the angle without jumps. This adjustment makes Fig. 7 clearer and prevents any misinterpretation related to data loss or anomalies.

5. As already mentioned I do not want to force you into an additional work. I genuinely believe that the two papers below are related to your work, so maybe you can find them useful for this paper or in a follow-up study.

Jia, Xiaolu, et al. "Experimental study on the evading behavior of individual pedestrians when confronting with an obstacle in a corridor." Physica A: Statistical Mechanics and its Applications 531 (2019): 121735.

Murakami, Hisashi, et al. "Spontaneous behavioral coordination between avoiding pedestrians requires mutual anticipation rather than mutual gaze." Iscience 25.11 (2022).

Thank you for your suggestion. As you mentioned, the recommended papers are highly relevant to our manuscript, so we cited them in the main text (Murakami et al., 2022 in Literature review; Jia et al., 2019 in Discussion). We appreciate your insightful feedback.

Once again, we appreciate the opportunity to revise our manuscript based on the reviewers’ valuable input. We look forward to hearing from you regarding our submission and will be happy to respond should you have any further questions or comments.

Sincerely,

Tatsuto Yamauchi, Hideki Tamura, Tetsuto Minami, and Shigeki Nakauchi

---

## [Editor Report · Decision Letter 1]

11 Feb 2025

PONE-D-24-41357R1Waist rotation angle as indicator of probable human collision-avoidance direction for autonomous mobile robotsPLOS ONE

Dear Dr. Tamura,

Thank you for submitting your manuscript to PLOS ONE. After careful consideration, we feel that it has merit but does not fully meet PLOS ONE’s publication criteria as it currently stands. Therefore, we invite you to submit a revised version of the manuscript that addresses the points raised during the review process.

**Some additional comments are given as follows:**1. Abstract: Readers cannot know that is your Experiment 2 and 3 so I suggest removing them. Also, since only experiments 2 and 3 are mentioned, one would wonder where it Experiment 1. Of course, that is clear in the manuscript, but abstract is read before the main text.2. Fig. 5: I found if difficult to understand and generally not standalone. Maybe you can consider providing a number of snapshots, like in the old animation movies. Or use colors to show the change in time. For example, as robot and person move toward each other the color gets brighter (black, dark gray, light gray, etc.). b is also not that clear. The angle of the arrow changes a little, making it difficult to understand. What about using a top view where a person is moving the waist? In c I found “slower” and “more comfortable” redundant and creating more confusion.3. Fig. 6: As already said, I found the reporting of the scale here redundant.4. Fig. 7: I believe you are using MATLAB. If so, why don’t you use unwrap to “join” the angles and avoid the jump from 0 to 360? You can have negative values and the graphs looks clearer. The jump you see looks like there is some data loss or something strange happened, while in reality is simply related to the definition of the angle.5. As already mentioned I do not want to force you into an additional work. I genuinely believe that the two papers below are related to your work, so maybe you can find them useful for this paper or in a follow-up study.Jia, Xiaolu, et al. "Experimental study on the evading behavior of individual pedestrians when confronting with an obstacle in a corridor." Physica A: Statistical Mechanics and its Applications 531 (2019): 121735.Murakami, Hisashi, et al. "Spontaneous behavioral coordination between avoiding pedestrians requires mutual anticipation rather than mutual gaze." Iscience 25.11 (2022).

We look forward to receiving your revised manuscript.

Kind regards,

silas onyango awuor, msc

Academic Editor

PLOS ONE

**Journal Requirements:**

**Additional Editor Comments:**

congratulation for the input otherwise I realized that the reviewer comments is not well answered kindly take you time and act on the below comments.

Some additional comments are given as follows:

1. Abstract: Readers cannot know that is your Experiment 2 and 3 so I suggest removing them. Also, since only experiments 2 and 3 are mentioned, one would wonder where it Experiment 1. Of course, that is clear in the manuscript, but abstract is read before the main text.

2. Fig. 5: I found if difficult to understand and generally not standalone. Maybe you can consider providing a number of snapshots, like in the old animation movies. Or use colors to show the change in time. For example, as robot and person move toward each other the color gets brighter (black, dark gray, light gray, etc.). b is also not that clear. The angle of the arrow changes a little, making it difficult to understand. What about using a top view where a person is moving the waist? In c I found “slower” and “more comfortable” redundant and creating more confusion.

3. Fig. 6: As already said, I found the reporting of the scale here redundant.

4. Fig. 7: I believe you are using MATLAB. If so, why don’t you use unwrap to “join” the angles and avoid the jump from 0 to 360? You can have negative values and the graphs looks clearer. The jump you see looks like there is some data loss or something strange happened, while in reality is simply related to the definition of the angle.

5. As already mentioned I do not want to force you into an additional work. I genuinely believe that the two papers below are related to your work, so maybe you can find them useful for this paper or in a follow-up study.

Jia, Xiaolu, et al. "Experimental study on the evading behavior of individual pedestrians when confronting with an obstacle in a corridor." Physica A: Statistical Mechanics and its Applications 531 (2019): 121735.

Murakami, Hisashi, et al. "Spontaneous behavioral coordination between avoiding pedestrians requires mutual anticipation rather than mutual gaze." Iscience 25.11 (2022).

---

## [Author Response · Author response to Decision Letter 2]

11 Feb 2025

Response Letter

congratulation for the input otherwise I realized that the reviewer comments is not well answered kindly take you time and act on the below comments.

Thank you for reaching out to us. As outlined below, we have made the necessary revisions. We kindly ask for your review and confirmation.

Some additional comments are given as follows:

1. Abstract: Readers cannot know that is your Experiment 2 and 3 so I suggest removing them. Also, since only experiments 2 and 3 are mentioned, one would wonder where it Experiment 1. Of course, that is clear in the manuscript, but abstract is read before the main text.

Thank you for your feedback. As you suggested, we have removed explicit references to Experiment 2 and Experiment 3 from the abstract to improve readability.

2. Fig. 5: I found if difficult to understand and generally not standalone. Maybe you can consider providing a number of snapshots, like in the old animation movies. Or use colors to show the change in time. For example, as robot and person move toward each other the color gets brighter (black, dark gray, light gray, etc.). b is also not that clear. The angle of the arrow changes a little, making it difficult to understand. What about using a top view where a person is moving the waist? In c I found “slower” and “more comfortable” redundant and creating more confusion.

Thank you for your suggestion. We have revised Fig. 5 as follows:

Fig. 5a: As you suggested, we illustrated the progression from the beginning to the end of a trial using snapshots from a video.

Fig. 5b: We clarified the movement of the participants’ waist by using a schematic top-down view, showing the changes in the waist angle (θ) for each condition (Beginning, Early, Same, Late).

Fig. 5c: We simplified the labels to improve readability.

3. Fig. 6: As already said, I found the reporting of the scale here redundant.

We have revised the y-axis label and caption in Figs. 6 and 8.

4. Fig. 7: I believe you are using MATLAB. If so, why don’t you use unwrap to “join” the angles and avoid the jump from 0 to 360? You can have negative values and the graphs looks clearer. The jump you see looks like there is some data loss or something strange happened, while in reality is simply related to the definition of the angle.

Thank you for your suggestion. We applied the unwrap function in MATLAB to ensure a continuous representation of the angle without jumps. This adjustment makes Fig. 7 clearer and prevents any misinterpretation related to data loss or anomalies.

5. As already mentioned I do not want to force you into an additional work. I genuinely believe that the two papers below are related to your work, so maybe you can find them useful for this paper or in a follow-up study.

Jia, Xiaolu, et al. "Experimental study on the evading behavior of individual pedestrians when confronting with an obstacle in a corridor." Physica A: Statistical Mechanics and its Applications 531 (2019): 121735.

Murakami, Hisashi, et al. "Spontaneous behavioral coordination between avoiding pedestrians requires mutual anticipation rather than mutual gaze." Iscience 25.11 (2022).

Thank you for your suggestion. As you mentioned, the recommended papers are highly relevant to our manuscript, so we cited them in the main text (Murakami et al., 2022 in Literature review; Jia et al., 2019 in Discussion). We appreciate your insightful feedback.

Once again, we appreciate the opportunity to revise our manuscript based on the reviewers’ valuable input. We look forward to hearing from you regarding our submission and will be happy to respond should you have any further questions or comments.

Sincerely,

Tatsuto Yamauchi, Hideki Tamura, Tetsuto Minami, and Shigeki Nakauchi

---

## [Decision Letter · Decision Letter 2]

6 Apr 2025

PONE-D-24-41357R2Waist rotation angle as indicator of probable human collision-avoidance direction for autonomous mobile robotsPLOS ONE

Dear Dr. Tamura,

Thank you for submitting your manuscript to PLOS ONE. The reviewer who evaluated your manuscript has deemed it suitable for publication but I noticed two minor issues that I would like you to consider before an Accept desicion can be made.  Therefore, we invite you to submit a revised version of the manuscript that addresses the points raised during the review process.

We look forward to receiving your revised manuscript.

Kind regards,

Dimitris Voudouris

Academic Editor

PLOS ONE

**Journal Requirements:**

**Additional Editor Comments:**

The term PSE (e.g., line 264 and elsewhere) is not appropriate as the term "subjective equality" is not well aligned with your analysis. I recommend revising it to something else. Given that your x-axis in Figure 3B indicats the robot's starting angle, you could use this term instead.

Lines 316-317, consider using numbers ("reached 1 and 1.5 times the average range..").

Reviewers' comments:

Reviewer's Responses to Questions

**Comments to the Author**

1. If the authors have adequately addressed your comments raised in a previous round of review and you feel that this manuscript is now acceptable for publication, you may indicate that here to bypass the “Comments to the Author” section, enter your conflict of interest statement in the “Confidential to Editor” section, and submit your "Accept" recommendation.

Reviewer #1: All comments have been addressed

2. Is the manuscript technically sound, and do the data support the conclusions?

Reviewer #1: Yes

3. Has the statistical analysis been performed appropriately and rigorously? 

Reviewer #1: Yes

4. Have the authors made all data underlying the findings in their manuscript fully available?

Reviewer #1: Yes

5. Is the manuscript presented in an intelligible fashion and written in standard English?

Reviewer #1: Yes

6. Review Comments to the Author

**Reviewer #1:**  I do not have further comments and I believe the manuscript can be accepted for publication in the current form.

7. PLOS authors have the option to publish the peer review history of their article (what does this mean? ). If published, this will include your full peer review and any attached files.

**Do you want your identity to be public for this peer review?** For information about this choice, including consent withdrawal, please see our Privacy Policy .

Reviewer #1: No

---

## [Author Response · Author response to Decision Letter 3]

8 Apr 2025

Response Letter

The term PSE (e.g., line 264 and elsewhere) is not appropriate as the term "subjective equality" is not well aligned with your analysis. I recommend revising it to something else. Given that your x-axis in Figure 3B indicats the robot's starting angle, you could use this term instead.

Thank you for pointing this out. We agree that the term “PSE” may not precisely reflect the nature of our analysis. In response to your suggestion, we have replaced it with either “starting angle” or “estimated starting angle,” depending on the context throughout the manuscript, including the caption of Figure 3B and the relevant sections in the main text.

Lines 316–317, consider using numbers ("reached 1 and 1.5 times the average range..").

Thank you for your suggestion. We have revised the sentence to use numeric expressions for clarity.

“In the Early and Late conditions, the AMR initiated deflection when the participant’s waist rotation angle reached 0.5 and 1.5 times the average range of waist rotation angles, respectively.”

---

## [Editor Report · Decision Letter 3]

11 Apr 2025

Waist rotation angle as indicator of probable human collision-avoidance direction for autonomous mobile robots

PONE-D-24-41357R3

Dear Dr. Tamura,

We’re pleased to inform you that your manuscript has been judged scientifically suitable for publication and will be formally accepted for publication once it meets all outstanding technical requirements.

Kind regards,

Dimitris Voudouris

Academic Editor

PLOS ONE
---

## [Editor Report · Acceptance letter]

PONE-D-24-41357R3

PLOS ONE

Dear Dr. Tamura,

I'm pleased to inform you that your manuscript has been deemed suitable for publication in PLOS ONE. Congratulations! Your manuscript is now being handed over to our production team.

Kind regards,

on behalf of

Dr. Dimitris Voudouris

Academic Editor

PLOS ONE